# Enhancing Vision-Language Model Pre-training with Image-text Pair Pruning Based on Word Frequency

## Abstract

We propose **W**ord-**F**requency-based Image-Text **P**air **P**runing (WFPP), a novel data pruning method that improves the efficiency of VLMs. Unlike MetaCLIP, our method does not need metadata for pruning, but selects text-image pairs to prune based on the content of the text. Specifically, WFPP prunes text-image pairs containing high-frequency words across the entire training dataset. The effect of WFPP is to reduce the dominance of frequent words. The result a better balanced word-frequency distribution in the dataset, which is known to improve the training of word embedding models. After pre-training on the pruned subset, we fine-tuned the model on the entire dataset for one additional epoch to achieve better performance. Our experiments demonstrate that applying WFPP when training a CLIP model improves performance on a wide range of downstream tasks. WFPP also provides the advantage of speeding up pre-training by using fewer samples. Additionally, we analyze the training data before and after pruning to visualize how WFPP changes the balance of word frequencies. We hope our work encourages researchers to consider the distribution of words in the training data when pre-training VLMs, not limited to CLIP.

## 1 Introduction

Large-scale pre-trained Vision-Language Models (VLMs) are gaining popularity, because of their remarkable zero-shot transferability (Radford et al., 2021; Li et al., 2023b; Zhai et al., 2022; Jia et al., 2021; Kim et al., 2021). This makes the VLM a foundational model with wide applicability in a variety of downstream tasks (Rombach et al., 2022; Ramesh et al., 2022). The success of VLMs rests on two key points: (1) Large-scale image-text pair datasets crawled from the Internet (Sharma et al., 2018; Changpinyo et al., 2021; Thomee et al., 2016; Schuhmann et al., 2022). (2) Large-scale transformers used as image and text encoder (Dosovitskiy et al., 2021; Vaswani et al., 2017).

Despite the importance of the scale of large-scale datasets, previous work has shown that pruning the training dataset can lead to improvements. MetaCLIP (Xu et al., 2024) employs a strategy that leverages metadata associated with text-image pairs in order to create a subset of CLIP training data. MetaCLIP pruning results in a new dataset that is balanced at the level of metadata categories (called "entries"), such that the number of texts associated with any given category does not exceed a threshold. The improvements of pruning are accompanied by the computational speed-up that results when the size of the training dataset is reduced.

In this paper, we propose that the decision to prune an image-text pair should be based directly on information about the frequency of words in that pair. Our method, **W**ord-**F**requency-based Image-Text **P**air **P**runing (WFPP) is inspired by observations of the importance of balancing word frequencies for training word embedding models. Specifically, Mikolov et al. (2013) introduced a technique that subsamples frequent words in text data, in order to speed up the training and enhance the quality of word representations. Like Mikolov et al. (2013), we consider a dataset to be better balanced in terms of word-frequency when the frequencies of frequent words are less dramatically higher than the frequencies of infrequent words. Our idea is also consistent with work that has demonstrated that neural networks tend to learn more from the majority class due to the higher number of examples available (Ross & Dollár, 2017; Buda et al., 2018).

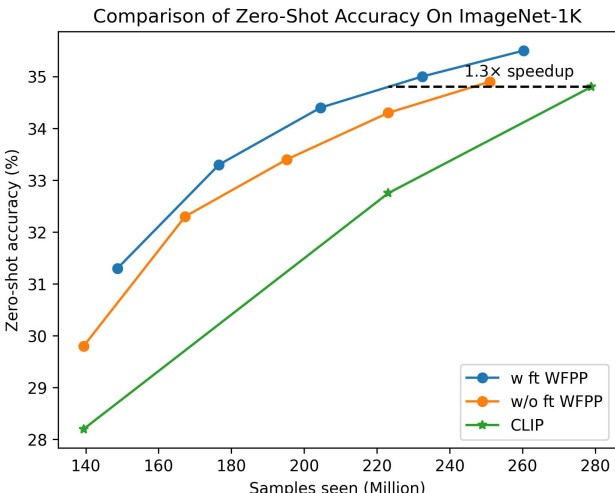

Figure 1: **Zero-shot accuracy on ImageNet-1K classification.** CLIP is trained on the CC12M dataset (Changpinyo et al., 2021). Using our Word-Frequency-based Image-Text Pair Pruning (WFPP), we achieve comparable performance, while using only approximately 77% of the image-text pairs (1.3× speedup). The image encoder is ViT-B-16 (Dosovitskiy et al., 2021). The "ft" is fine-tuning. The w/o ft is without fine-tuning. "Samples seen" refers to the number of samples processed during pre-training.

WFPP uses a simple yet effective text-level score based on word probabilities to prune image-text pairs from the data set in which the text contains frequent words. Word balance could also be improved by removing individual words from the training data, such as proposed by Liang & Larson (2023). However, such an approach does not remove images, which is disadvantageous since the image encoder usually accounts for a large portion of the training time. WFPP in contrast selects entire image-text pairs for removal. Note that the WFPP manner of removing texts does not impact the overall vocabulary richness, measured in vocabulary size, which is important to maintain.

Figure 1 demonstrates that our method effectively speeds up zero-shot classification tasks by 1.3× while maintaining the performance of CLIP trained on the full training set. Moreover, our findings demonstrate that WFPP outperforms CLIP in a variety of downstream tasks, including zero-shot classification and zero-shot image-text retrieval across multiple datasets.

WFPP offers two advantages over the data pruning proposed by MetaCLIP. First, WFPP selects image-text pairs on the basis of an individual score. In contrast, MetaCLIP selects image-text pairs on the basis of the metadata category they are associated with, meaning that the selection process is less specific. Second, WFPP selects image-text pairs directly using the content of the training data and without the need for a list of metadata categories. Often, the data collection process for training sets involves the use of queries, which can be adopted as the metadata categories for the training examples, as in CLIP. However, it is not a given that the dataset was created in this way. Further, the set of categories use to collect one dataset might not be optimal to subsample another dataset, e.g., with a different topical distribution. The contributions of this work can be summarized as follows:

- Image-text-pair-level pruning: We introduce an image-text pair pruning method based on word frequency, which substantially reduces the computational requirements while pre-training VLMs without compromising performance.

- Word balance: Our approach contributes to the evidence on the importance of word balance, the importance of good design decisions for large-scale datasets, rather than just scaling them up.

- We provide extensive experiments and analyses that CLIP trained on a dataset sampled with our approach outperforms CLIP trained on an unsampled dataset.

Our code is available online[1].

## 2 RELATED WORK

Due to its remarkable zero-shot transferability, the visual-semantic embedding model as a foundational model has received sustained attention from researchers. In this section, we describe the pre-training methods for vision-language models and discuss related works to accelerate their pre-training.

**Vision-Language Models:** DeViSE (Frome et al., 2013) learns visual-semantic embedding from labeled embeddings which are generated from pre-trained skip-gram on 5.7 million documents (5.4 billion words) extracted from *wikipedia.org*. The semantic knowledge learned from language provides the zero-shot prediction capability of a visual model, which improves performance on unseen data. To enhance visual-semantic embeddings and achieve good zero-shot performance, CLIP and ALIGN scale the data to 400M to learn the better visual-semantic embedding that achieves remarkable zero-shot performance across 27 datasets (Radford et al., 2021; Jia et al., 2021). These models are pre-trained by contrastive learning, which pushes positive image-text pairs closer to each other and separates negative image-text pairs, aligning the vision and language by acquiring a visual-semantic embedding from the natural language supervision. However, pre-training VLMs on large-scale data are quite expensive, demanding thousands of GPU days (Cherti et al., 2023; Radford et al., 2021).

**Efficient Language-Image Pre-training for CLIP:** Several methods have been proposed to enhance the efficiency of CLIP models. DeCLIP Li et al. (2022b) leverages additional self-supervised losses to improve image and text representations which extends CLIP by incorporating intra-modal self-supervision, cross-modal multi-view supervision, and nearest-neighbor supervision. FILIP Yao et al. (2022) introduces a cross-modal late interaction module that refines the contrastive objective by using token-wise maximum similarity between visual and textual tokens. UniCLIP Lee et al. (2022) unifies inter-domain (image-text) and intra-domain (image-image, text-text) contrastive losses into a single universal embedding space, capturing comprehensive relationships across and within modalities. Knowledge distillation approaches such as TinyCLIP Wu et al. (2023) and MoPE-CLIP Lin et al. (2024) transfer knowledge from large pre-trained models to smaller ones via cross-modal distillation to reduce the pre-training budget. Additionally, methods like MobileCLIP Vasu et al. (2024) and ALIP Yang et al. (2023) generate synthetic captions using models pre-trained on large datasets (CoCa Yu et al. (2022) and OFA_base Wang et al. (2022), respectively) to improve CLIP pre-training. Although these proposed pre-training strategies and synthetic caption methods are efficient, they may still exhibit imbalanced word distributions of pre-training data. Applying WFPP to these methods can further improve training efficiency. Moreover, Fast Language-Image Pre-training (FLIP) (Li et al., 2023b) removes a large portion of image patches to speed up pre-training VLMs. FLIP uses ViT as an image encoder, reducing computation by 2-4× by removing 50%-75% patches of the image while obtaining better accuracy than the unmasked model (Li et al., 2023b). In addition, they randomly masked 50% of the text to pre-train the VLMs, However, this approach does not work well in text encoders, and the performance of zero-shot classification on ImageNet is decreased. Resource-efficient CLIP (RECLIP) (Li et al., 2023a) employs a smaller version of images for the initial pre-training of CLIP and subsequently fine-tunes the models using larger versions of the images. The pre-training RECLIP with an image size of 64 × 64 reduces compute resource usage by approximately 80% while still outperforming CLIP on image-text retrieval tasks (Li et al., 2023a). Subsampling of Frequent Words for Contrastive Language-Image Pre-training (SW-CLIP) (Liang & Larson, 2023) proposed a frequency-based word subsampling technique to reduce text length by half for pre-training VLMs, but does not remove image-text pairs.

**Data Puning in VLMs:** Several data pruning methods have been developed for Natural Language Processing Sorscher et al. (2022); Marion et al. (2023) and Computer Vision Tan et al. (2024). However, in this work, we focus on multimedia data pruning. Metadata-Curated Language-Image Pre-training (MetaCLIP) (Xu et al., 2024) creates a balanced subset based on the metadata distribution to pre-train VLMs. To balance the training data, MetaCLIP selects image-text pairs from the data pool where the text contains a metadata entry. These metadata entries consist of four components: WordNet synsets, Wiki unigram, Wiki bigram, and Wiki titles. MetaCLIP utilizes a rich metadata source with 500k entries covering a wide range of concepts. However, it only seeks a balance at the level of the entries (metadata categories), which could lead to certain words being under or

---

[1] https://anonymous.4open.science/r/WFPP-1656/

Table 1: Two example texts. In the probability row, we show the probability of removing words in the text. Then, the last column is the probability of a text being removed from the data; it is the joint probability of the words in the text. The value of $t$ in Equation 2 is set to $10^{-7}$.

| text | a | picture | of | barcode | $S(t_j)$ |
|---|---|---|---|---|---|
| $f(w_i)$ | 0.9980 | 0.9861 | 0.9978 | 0.8342 | 0.20479 |
| text | a | picture | of | dog | |
| $f(w_i)$ | 0.9980 | 0.9861 | 0.9978 | 0.9878 | 0.24249 |

over-represented in the sampled training data. In this paper, we prune image-text pairs based on word frequency to create a more balanced subset for pre-training VLMs. Our approach is also easier to implement, as it does not require collecting and filtering thousands of entries or engaging in complex, time-consuming curation processes.

## 3 METHOD

In this section, we present WFPP, a method designed to enhance the pre-training of Vision-Language Models (VLMs) by strategically selecting image-text data from the dataset based on word frequency. Following the data pruning process, we can effectively pre-train the VLM using a reduced portion of the dataset without compromising its performance.

Following our proposed principles for building more balanced data, we remove as much text as possible from the dataset that contains higher-frequency words. The removal probability of a text is defined by the joint probability of the words in the text. This approach maintains the diversity of the data when filtering the information, as well as maintaining a balance between frequent and infrequent words.

To achieve our goal, we first compute the frequency of words using the equation:

$$f(w_i) = \frac{c(w_i)}{\sum_{i=1}^{n} c(w_i)} \tag{1}$$

In this equation, $f(w_i)$ represents the frequency of the word $w_i$, and $c(w_i)$ stands for the word count for $w_i$. Next, we determine the probability of a word being discarded according to Eq. 2:

$$P(w_i) = \begin{cases} 1 - \sqrt{\frac{t}{f(w_i)}} & f(w_i) > t \\ 1 & \text{otherwise} \end{cases} \tag{2}$$

In this equation, $t$ serves as the threshold controlling the probability of a word being discarded (Mikolov et al., 2013). We set $P(w_i)$ to 1 if $f(w_i) <= t$, in this way, very rare words do not affect the probability of the text being discarded. Due to the limited number of rare words occurring in the dataset, the impact on model performance is very slight.

Lastly, we calculate the joint probability of a text being discarded from the dataset according to Eq. 3:

$$S(t_j) = \frac{1}{n} \prod_{i=1}^{n} P(w_i) \tag{3}$$

In this equation, $t_j$ represents the $j - th$ text, while $S(t_j)$ represents both the joint probability of words being discarded and the probability of the text being discarded from the dataset. $n$ is the length of the text, and the maximum value of $n$ is equal to the maximum value of the input text. The advantage of this equation is that it filters out text containing frequent words to create a subset of the dataset with a balanced word distribution.

In Table 1, the rows display the probabilities of text being discarded from the dataset. On one hand, text containing infrequent words has a low probability of being discarded from the dataset (row

Table 2: The details of pre-training and fine-tuning setup.

| Configuration | Pre-training Value | Fine-tuning Value |
|---|---|---|
| Optimizer | AdamW (Loshchilov & Hutter, 2019) | AdamW (Loshchilov & Hutter, 2019) |
| Learning rate | 1e-3 | 1e-5 |
| Weight decay | 0.1 | 0.05 |
| Optimizer momentum | $\beta 1, \beta 2 = 0.9, 0.999$ (Chen et al., 2020) | $\beta 1, \beta 2 = 0.9, 0.999$ (Chen et al., 2020) |
| Learning rate schedule | Cosine decay (Loshchilov & Hutter, 2017) | Cosine decay (Loshchilov & Hutter, 2017) |
| Warmup steps | 10k | 10% |
| Epochs | 30 | 1 |
| Numerical precision | Automatic mixed precision | Automatic mixed precision |
| Augmentation | RandomResizedCrop | RandomResizedCrop |

1). On the other hand, texts containing frequent words, as illustrated in the second row of Table 1, have a high probability of being discarded from the dataset. After pruning the data based on word frequency, we obtain a new balanced dataset that maintains the diversity of the data while reducing the training samples. Subsequently, we sort the texts by their joint probability $S(t_i)$ and select a specific number of samples in order to pre-train the model. We pre-trained the model using different sampling proportions to identify the proportion of data that achieves comparable performance to the model pre-trained on the entire dataset.

## 4 EXPERIMENTS

### 4.1 IMPLEMENTATION DETAILS

**Dataset** In our experiments, we utilize CC3M and CC12M to pre-train our model which includes about 3M and 12M image-text pairs (Sharma et al., 2018; Changpinyo et al., 2021). These datasets were chosen because they collect a large number of different image-text pairs, providing diverse content for pre-training effective VLMs. We employed various methods for subsampling the dataset, including random selection, and frequency-based sampling. This comparative analysis aims to illustrate the advantage of more diverse data over less diverse data in the context of pre-training models. As a result, we have successfully downloaded 2.72 million data items for CC3M, and 9.30 million data items for CC12M (Sharma et al., 2018; Changpinyo et al., 2021). In these datasets, each image has an associated text. We also use the COCO (Lin et al., 2014) and Flick30K (Young et al., 2014) to evaluate the zero-shot retrieval performance, and in these datasets, each image has five associated texts to describe the context of the image.

**Architecture** For the image encoder, we used ViT-B-16 (Dosovitskiy et al., 2021) to encode the image and the input size of the image is 224. We use a Transformer-based model (Vaswani et al., 2017) as the text encoder and the text length is 32 (Li et al., 2023b). Following CLIP and OpenCLIP (Radford et al., 2021; Cherti et al., 2023), we compute the similarity score based on the cosine similarity between image and text embeddings. The model is pre-trained by InfoNCE loss (Oord et al., 2018), and the similarity scores are scaled by a learnable temperature parameter (Radford et al., 2021).

**Training and Fine-tuning** We first pre-trained the model on the entire dataset, as well as on random and WFPP subsets, for 30 epochs. Then, we fine-tuned the models pre-trained on the subsets using the entire dataset for an additional epoch. This additional epoch of fine-tuning aims to bridge the distribution gap between the pre-training and inference stages and to account for any unknown concepts that may have been present in the initially removed data. The value of $t$ in Eq. 2 is set to $10^{-7}$, and the details of pre-training and fine-tuning configuration are shown in Table 2.

### 4.2 EVALUATION

To evaluate the zero-shot classification performance on ImageNet, where the model correctly classifies data into never-before-seen categories during training. We follow the prompt engineering of CLIP and OpenCLIP (Radford et al., 2021; Cherti et al., 2023), utilizing their codebase, which includes a set of 80 templates. We then calculated the cosine similarity score between the image and text embeddings to evaluate the correspondence.

Table 3: **Zero-shot classification accuracy on ImageNet-1K.** We pre-train models by sampling image-text pairs sorted according to Equation 3 at sampling rates ranging from 50% to 90%. The pre-training dataset is **CC12M** (Changpinyo et al., 2021), and the image encoder used is ViT-B-16 (Dosovitskiy et al., 2021). "Samples seen" refers to the proportion of the dataset processed during pre-training, with 100% set as 1.00. w/ft and w/o/ft is with and without fine-tuning.

| Method | Sample Size | w/o/ft | w/ft | Samples seen (w/ft) |
|--------|-------------|--------|------|---------------------|
| CLIP | 9.30M | 34.8 | ✗ | $1.00\times$ |
|  | 4.65M (50%) | 28.2 | 30.2 | $0.53\times$ |
| WFPP | 4.65M (50%) | 29.8 | 31.3 | $0.53\times$ |
|  | 5.58M (60%) | 32.3 | 33.3 | $0.63\times$ |
|  | 6.51M (70%) | 33.4 | 34.4 | $0.73\times$ |
|  | 7.44M (80%) | 34.3 | 35.0 | $0.83\times$ |
|  | 8.37M (90%) | **34.9** | **35.5** | $0.93\times$ |

Table 4: **Zero-shot robustness evaluation**, Comparison of zero-shot accuracy performance between CLIP trained on the original datasets and on data pruned with WFPP on various classification benchmarks.

| Dataset | CLIP | WFPP | | | | |
|---------|------|------|------|------|------|------|
|  |  | 50% | 60% | 70% | 80% | 90% |
| ImageNet-A | 7.69 | 6.71 | 7.79 | 7.49 | 8.11 | **8.15** |
| ImageNet-O | 38.05 | 35.80 | 36.85 | 36.70 | **39.15** | 38.45 |
| ImageNet-R | **45.02** | 35.94 | 38.97 | 41.32 | 43.43 | 44.16 |
| ImageNet Sketch | **22.89** | 17.17 | 18.82 | 20.98 | 21.72 | 22.53 |
| ImageNetV2 | 30.15 | 26.44 | 28.10 | 29.72 | 30.41 | **30.90** |
| ObjectNet | 20.73 | 19.21 | 20.91 | 21.30 | 21.83 | **21.94** |
| Average | 27.42 | 23.55 | 25.24 | 26.25 | 27.44 | **27.69** |

**Zero-shot ImageNet Classification.** First of all, as shown in Figure 1, we pre-train the model on different size subsets of CC12M. When we evaluate our method on zero-shot accuracy on ImageNet-1K (Deng et al., 2009) validation, we only need 80% of the computation to achieve a better performance as the CLIP counterpart. Specifically, as detailed in the first (100%) and sixth rows (80%) of Table 3, our model, utilizing just 83.3% of the computational resources compared to the model trained on the full dataset, achieves better performance in the zero-shot ImageNet-1K classification task, with scores of 35.0% versus 34.8%.

As shown in the second and third rows of Table 3, the model pre-trained on a subset pruned using a word frequency-based method performs significantly better in the zero-shot image classification task than the model trained with a randomly pruned subset (29.8% vs. 28.2%). The WFPP method outperforms the random method by 1.7% before fine-tuning. After fine-tuning for an additional epoch on the entire dataset, our method continues to outperform the random method by 1.1%. Notably, the differences between the random and frequency-based methods become smaller after fine-tuning the model on the entire dataset.

**Subsampling more data.** As demonstrated in Table 3, subsampling 90% of the image-text pairs from the CC12M dataset allows us to attain comparable performance without needing to fine-tune the model on the entire dataset. This observation suggests that excluding the 10% of image-text pairs containing frequent words results in only a slight performance degradation of 0.1%. After fine-tuning the model on the entire dataset, the model trained on WFPP-pruned data outperforms the original CLIP by 0.7% on ImageNet-1K. Moreover, increasing the subsampling percentage from 50% to 60%, 70%, 80%, and 90% results in performance improvements of 2.5%, 1.1%, 0.9%, and 0.6%, respectively. This indicates that data efficiency decreases as the amount of data increases. The latter part of the data contains more high-frequency words than the former part of the data. Blindly adding data becomes increasingly inefficient. As a result, the efficiency of adding more data indiscriminately decreases over time. For the reason, we do not recommend to use the latter part of the data when using WFPP data pruning.

Table 5: **Zero-shot accuracy on more classification datasets.** Comparison of zero-shot classification accuracy between CLIP trained on the original datasets and on data pruned with WFPP on various classification benchmarks.

| Dataset | CLIP | WFPP | | | | |
|---|---|---|---|---|---|---|
| | | 50% | 60% | 70% | 80% | 90% |
| Food-101 | 40.57 | 38.64 | 40.77 | 41.22 | 42.30 | **43.25** |
| CIFAR-10 | 63.28 | 52.44 | 57.55 | 63.57 | 65.60 | **66.88** |
| CIFAR-100 | 32.25 | 27.10 | 30.40 | **32.86** | 28.85 | 29.56 |
| CUB200 | 8.13 | 8.20 | 8.30 | 8.77 | **9.42** | 8.75 |
| SUN397 | 48.02 | 45.51 | 47.67 | 48.46 | 50.07 | **50.61** |
| Cars | 7.19 | 4.61 | 4.56 | 5.65 | **7.40** | 7.18 |
| Aircraft | 2.84 | 1.62 | 2.68 | 2.45 | 2.20 | **2.89** |
| DTD | 15.43 | 14.57 | 15.74 | 17.18 | 16.28 | **18.88** |
| OxfordPets | 56.11 | 45.98 | 53.27 | 56.38 | 53.58 | **56.49** |
| Caltech-101 | **70.16** | 64.56 | 66.54 | 68.01 | 68.93 | 69.23 |
| Kinetics700 | **24.23** | 21.96 | 22.91 | 23.41 | 24.11 | 24.05 |
| Flowers102 | 1.87 | **2.92** | 2.42 | 2.36 | 1.58 | 1.58 |
| MNIST | 9.49 | **18.08** | 14.29 | 14.54 | 11.30 | 15.13 |
| STL10 | 91.76 | 90.05 | 89.96 | 90.54 | 90.55 | **92.14** |
| EuroSAT | 22.00 | 17.66 | 25.72 | 22.56 | **26.14** | 24.32 |
| Resisc45 | 36.57 | 35.30 | 33.33 | 35.17 | 33.24 | **36.59** |
| GTSRB | 10.40 | 4.85 | 9.54 | 6.25 | **10.55** | 5.05 |
| KITTI | 36.43 | 36.83 | 34.89 | **39.04** | 34.89 | 37.30 |
| Country211 | 4.35 | 4.39 | 4.50 | 4.44 | 4.10 | **4.96** |
| PCAM | **52.89** | 52.55 | 52.76 | 52.39 | 51.67 | 52.62 |
| UCF101 | 38.14 | 35.74 | 37.96 | 39.02 | **41.42** | 39.02 |
| CLEVR | 18.27 | 17.74 | 13.23 | 19.14 | 16.55 | **25.00** |
| HatefulMemes | 52.62 | 53.26 | **54.30** | 50.09 | 50.79 | 51.53 |
| SST2 | 50.47 | 45.85 | 50.03 | 48.27 | **51.29** | 50.03 |
| ImageNet | 34.80 | 31.31 | 33.33 | 34.42 | 35.00 | **35.50** |
| **Average** | 33.13 | 30.87 | 32.27 | 33.05 | 33.11 | **33.94** |

**Zero-shot Robustness Evaluation** Following the methodology of CLIP (Radford et al., 2021), we evaluate robustness in Table 4. Using 80% of the image-text pairs, we achieve a comparable average performance to CLIP (27.44% vs. 27.42%) on these 6 datasets. Consequently, adding an additional 10% of image-text pairs results in only a 0.25% improvement, indicating that the efficiency of indiscriminately adding data with frequent words decreases over time in these datasets as well.

**Zero-shot Classification on More Datasets** ImageNet is a general-purpose dataset for evaluating the benchmark performance of VLMs, providing a benchmark for their effectiveness. While it provides a performance reference for VLMs, it's crucial to recognize that large-scale VLMs are destined for application across diverse datasets and tasks. Consequently, evaluating our approach to various datasets becomes imperative to underscore the significance of achieving a balance between data and diversity. The datasets featured in Table 5 hail from various domains, illustrating the advantages derived from this balanced approach to data diversity Specifically, within the CC12M dataset, WFPP demonstrates comparable average performance to CLIP in the zero-shot classification task across 26 datasets. First, the model is pre-trained on 80% of the data to achieve a similar performance as the model pre-trained on the entire dataset, which requires only 83.3% of CLIP's computational resources. Moreover, on the CC12M dataset—as shown in columns 2 and 4 of Table 5—WFPP requires only 63.3% of the computational resources compared to CLIP while achieving a similar average performance across 26 datasets (33.13% vs. 32.27%). Furthermore, when using 90% of the CC12M dataset, WFPP outperforms CLIP by an average of 0.83% across 26 datasets. The zero-shot classification performance of the majority of the datasets demonstrates improvement due to the diversity and balance in the data.

**Zero-shot retrieval** The task of image-text retrieval involves retrieving information from one modality (text or image) based on the given information from another modality (image or text). We evaluate WFPP for zero-shot image-text retrieval on COCO (Lin et al., 2014) and Flickr30K (Young et al.,

Table 6: **Zero-shot Image-Text Retrieval:** We evaluate CLIP's and WFPP image-text retrieval performance on both COCO and Flickr30k datasets.

| Model | Sample Size | Text Retrieval | | | | | | Image Retrieval | | | | | |
| | | Flickr30k | | | COCO | | | Flickr30k | | | COCO | | |
| | | R@1 | R@5 | R@10 | R@1 | R@5 | R@10 | R@1 | R@5 | R@10 | R@1 | R@5 | R@10 |
| CLIP | 9.30M | 59.17 | 85.40 | 90.24 | 34.32 | 61.14 | 72.50 | 45.38 | 71.95 | 81.05 | 22.65 | 46.94 | 58.86 |
| WFPP | 4.65M (50%) | 50.99 | 78.50 | 85.40 | 30.16 | 54.72 | 66.88 | 36.90 | 64.77 | 74.14 | 18.67 | 40.69 | 52.74 |
| | 5.58M (60%) | 57.00 | 81.56 | 88.07 | 31.88 | 57.70 | 69.38 | 39.47 | 67.50 | 76.82 | 19.79 | 43.04 | 54.92 |
| | 6.51M (70%) | 56.51 | 82.94 | 89.74 | 34.06 | 59.46 | 71.18 | 41.03 | 68.93 | 79.63 | 21.02 | 45.10 | 57.22 |
| | 7.44M (80%) | 59.47 | 85.11 | 90.34 | 33.98 | 60.88 | 72.06 | 42.56 | 70.12 | 79.82 | 22.18 | 46.36 | 58.48 |
| | 8.37M (90%) | 60.55 | 84.91 | 89.94 | 34.24 | 60.52 | 71.70 | 43.49 | 71.18 | 80.10 | 22.61 | 46.30 | 58.34 |

Table 7: **Zero-shot accuracy on ImageNet-1K classification.** Comparison of unpruned CLIP and pruning with WFPP and the Meta-CLIP method using the CC3M dataset.

| Method | Sample Size | w/o/ft | w/ft | Samples seen (w/ft) |
| --- | --- | --- | --- | --- |
| CLIP | 100% | **17.4** | ✗ | 1.00× |
| | 50% | 11.5 | 13.1 | 0.53× |
| WFPP | 50% | 13.4 | 15.1 | 0.53× |
| | 60% | 14.2 | 16.2 | 0.63× |
| | 70% | 15.9 | 17.4 | 0.74× |
| | 80% | 16.7 | **17.9** | 0.83× |
| | 90% | 16.9 | 17.5 | 0.93× |

Table 8: **Zero-shot accuracy on ImageNet-1K classification.** We compared WFPP with Meta-CLIP, both of them pre-trained on 50% of the CC3M dataset.

| Method | Sample Size | w/o/ft | w/ft |
| --- | --- | --- | --- |
| CLIP | 50% | 11.5 | 13.1 |
| MetaCLIP | 50% | 12.9 | 14.8 |
| WFPP | 50% | **13.4** | **15.1** |

2014) datasets. Following (Karpathy & Li, 2015), we utilize 1000 and 5000 test set images for evaluating performance zero-shot image-text retrieval on Flickr30K and COCO, respectively. Recall@K scores (where $K = 1, 5, 10$) are reported, representing the percentage of total test samples wherein the correct sample is present among the first K returned candidate samples.

Even when using fewer training samples, WFPP demonstrates competitive performance compared to the original CLIP model in zero-shot image-text retrieval tasks on the COCO and Flickr30K datasets. As the sample size for WFPP increases from 50% to 90% of CLIP's training data, its retrieval metrics steadily improve across both datasets. Notably, at 90% sample size, WFPP surpasses CLIP in text retrieval on Flickr30K, achieving an R@1 score of 60.55 compared to 59.17. These results suggest that WFPP achieves similar or superior performance to CLIP while being more data-efficient.

**Pre-training on Different Dataset.** We also pre-trained the model on the smaller dataset CC3M (Sharma et al., 2018). As shown in Table 7, the performance of the model is similar, with fine-tuning, the performance of subsampling 70% of the data is already the same as CLIP. Increasing the subsampling percentage from 70% to 80% results in a further performance increase of 0.8% without fine-tuning However, increasing the subsampling percentage from 80% to 90% results in only a 0.2% increase. This suggests that the effect of adding text containing high-frequency words is very insignificant. Notably, when we fine-tune the model pre-trained on the 90% subset, the performance is 0.4% lower than the model pre-trained on the 80% subset. Furthermore, WFPP outperforms MetaCLIP by 0.5% before fine-tuning and by 0.3% after fine-tuning.

## 4.3 COMPARISON WITH METACLIP

MetaCLIP also aims to create a balanced subset based on the metadata distribution. Therefore, we compare our method to MetaCLIP pre-trained on the CC3M dataset. Using the open-source code provided by MetaCLIP, we created a 50% training subset of CC3M. Table 8 presents the zero-shot ImageNet-1K classification accuracy for CLIP, MetaCLIP, and WFPP. Without fine-tuning, MetaCLIP achieves an accuracy of 12.9%, outperforming the baseline CLIP's 11.5%. WFPP further improves this result to 13.4%, surpassing MetaCLIP by 0.5%. Moreover, with fine-tuning, WFPP attains the highest accuracy of 15.1%, exceeding both MetaCLIP (14.8%) and CLIP (13.1%). These findings indicate that WFPP not only enhances performance over the baseline CLIP but also outperforms MetaCLIP in zero-shot ImageNet-1K classification tasks.

Table 9: **Zero-shot accuracy on ImageNet-1K classification.** We sorted the texts according to Eq. 3 and pre-trained the model separately on the first and second halves of the data. The pre-training dataset is **CC12M**. "Samples seen" refers to the number of samples processed during pre-training.

| Method | Subsampling | Sample Size | w/o ft | w/ft | Samples seen (w/ ft) |
|---|---|---|---|---|---|
| CLIP | Random | | 28.2 | 30.2 | 0.53× |
| WFPP-First | First half | 4.65M (50%) | **29.8** | **31.3** | 0.53× |
| WFPP-Second | Second half | | 21.3 | 24.3 | 0.53× |

Table 10: **Zero-shot accuracy on ImageNet-1K classification.** We selected image-text pairs based on text length and compared the results with the random, length-based, and WFPP methods. The dataset is **CC12M** and the image encoder is ViT-B-16. w/ft and w/o/ft is with and without fine-tuning.

| Method | Subsampling | Sample Size | w/o/ft | w/ft | Samples seen (w/ft) |
|---|---|---|---|---|---|
| CLIP | random | 4.65M (50%) | 28.2 | 30.2 | 0.53× |
| | length-based | 4.65M (50%) | 22.6 | 27.8 | 0.53× |
| WFPP | frequency-based | 4.65M (50%) | 29.8 | 31.3 | 0.53× |

## 4.4 IMPACT OF WORD FREQUENCY DISTRIBUTION

To investigate the benefits of pre-training with a better-balanced word distribution, we experiment with a badly-balanced word distribution, as a contrast. Specifically, we sort CC12M by Eq. 3 and pre-train on the second half, which is more likely to contain high-frequency words compared to the first half, usually used by WFPP. As shown in Table 9, the model pre-trained on the first half of the subset outperforms the model pre-trained on the second half by 8.7%. After fine-tuning, WFPP-First still maintains a 7.0% advantage. In addition, WFPP-Second performs 6.9% worse than a model pre-trained on a randomly selected 50% subset of CC12M. This result confirms the importance of selecting texts in a way that balances word frequency by reducing the frequency of high-frequency words.

## 4.5 IMPACT OF TEXT LENGTH AND TEXT LENGTH NORMALIZATION

Equation 3 is normalized by text length, indicating that length also affects our method. To examine the impact of text length, we pruned the dataset based on text length, retaining only the longer texts. As shown in Table 10, pruning based on text length resulted in poorer performance compared to both the random method (22.5 vs. 28.2) and the frequency-based method (22.5 vs. 29.8).

Additionally, Equation 3 without length normalization tends to retain longer texts. Therefore, we removed text length normalization in Equation 3 to evaluate its impact on model performance. The revised equation used to sort the text in the dataset is shown below:

$$S(t_j) = \prod_{i=1}^{n} P(w_i) \qquad (4)$$

As shown in Table 11, the length normalization operation improves the zero-shot classification accuracy on ImageNet-1K for models pre-trained on the CC3M and CC12M datasets by 1.2% and 2.6%, respectively. After fine-tuning, the improvements remain at 0.8% and 0.4%, respectively.

## 4.6 DATA ANALYSIS

To gain insight into the nature of the impact of WFPP on word-frequency distribution, we visualize word-frequency distribution before and after pruning in Figure 2. The figure reveals that high-frequency words are removed at a higher rate compared to low-frequency words for WFPP. In random pruning, each word has about a 50% chance of being pruned. In contrast, under WFPP, the retention probabilities for most of these top 50 high-frequency words are lower than 50%, especially for words like "person", "illustration", and "background", whose retention rates are 34.59%, 22.81%, and 22.31%, respectively. For infrequent words not shown in the figure, such as "connector", "swords", and "grille", the retention rates are higher—84.53%, 55.55%, and 70.35%, respectively.

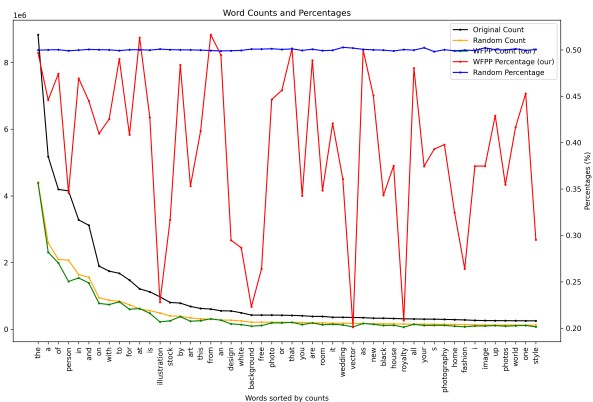

Figure 2: **Word Distribution:** The top-50 words in CC12M (Changpinyo et al., 2021) are shown after pruning 50% of image-text pairs using Random (orange) and WFPP (green) methods, and before pruning (black). We then calculate the word percentages for Random and WFPP before and after pruning. Words are ordered by frequency before pruning. The left Y-axis is the number of words and the left Y-axis is the percentage of words which is the number of words before data pruning divided by the number of words after data pruning.

Table 11: **Zero-shot accuracy on ImageNet-1K classification:** We removed length normalization from Equation 3 to observe the improvement that length normalization contributes.

| Dataset | Method | w/o/ft | w/ft |
|---------|--------|--------|------|
| CC3M | w/o normalization | 12.6 | 14.3 |
|  | w normalization | 13.4 | 15.1 |
| CC12M | w/o normalization | 27.2 | 29.9 |
|  | w normalization | 29.8 | 31.3 |

Table 12: **Vocabulary Size Comparison:** Comparison of vocabulary size before and after applying WFPP. WFPP reduced the number of image-text pairs in the CC12M dataset by 50%.

| Word Frequency | CLIP (100%) | WFPP (50%) |
|---------------|-------------|------------|
| More than 5 occurrences | 124,323 | 99,923 |
| More than 100 occurrences | 33,872 | 32,476 |

## 4.7 VOCABULARY SIZE ANALYSIS

To demonstrate that WFPP maintains the vocabulary diversity while improving the word-frequency balance, we calculated the vocabulary size before and after applying WFPP. As shown in Table 12, removing 50% of the image-text pairs, the number of vocabulary words with more than 100 occurrences has not decreased substantially. However, the number of words with frequencies between 5 and 100 decreases. Future work should investigate the impact of these words, which might be low, given their low frequencies.

## 5 CONCLUSION AND OUTLOOK

In this paper, we introduce WFPP, a novel data-pruning method to enhance vision-language pre-training. By pruning image-text pairs based on word frequencies in the corpus, we reduce the size of the training dataset reducing the necessary computation to pre-train the model. We demonstrate that pruning improves the word-frequency balance and we claim that this is the reason it is possible to prune data without impacting performance. In fact, across a wide range of tasks and datasets, our experiments demonstrate that after data pruning with WFPP, CLIP is actually able to achieve better performance that CLIP trained on unpruned data. WFPP also outperforms MetaCLIP pruning, which similarly aims to yield a balanced subset over the metadata distribution.

Moving forward, using WFPP to prune very large-scale datasets such as LAION-400M (Schuhmann et al., 2021) would be an interesting direction to explore. As shown in Figure 1, the improvement in zero-shot classification accuracy on ImageNet-1K increases substantially when 60% instead of 50% of CC12M data is retained after pruning. When more than 60% is retained, the rate of improvement falls off. We believe that between 50% to 60%, we are observing the linear scaling law demonstrated in CLIP (Radford et al., 2021; Cherti et al., 2023), but after that CC12M offers insufficient image-text pairs containing low-frequency words in order to maintain this rate of improvement. If we are right, it means that starting with a larger data set like LAION-400M, we could improve zero-shot classification accuracy would already exceed 40% with 280 million. The broader implication is that WFPP would maintain and possibly enhance its ability to improve performance as size of the dataset before pruning is increased. It would also be worthwhile to investigate alternative methods for pruning image-text pair data, such as term frequency-inverse document frequency (TF-IDF).

## 6 REPRODUCIBILITY STATEMENT

Our work builds upon open_clip[2], which provides detailed usage instructions, including how to download the dataset and pre-train the model. Because our experiments use datasets that are not too large, our results are broadly reproducible, i.e., using typical resources available in academic settings. Additionally, we have described our set up in detailed as also made all source code available, including scripts for data preprocessing, training, and evaluation, at WFPP [3].

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

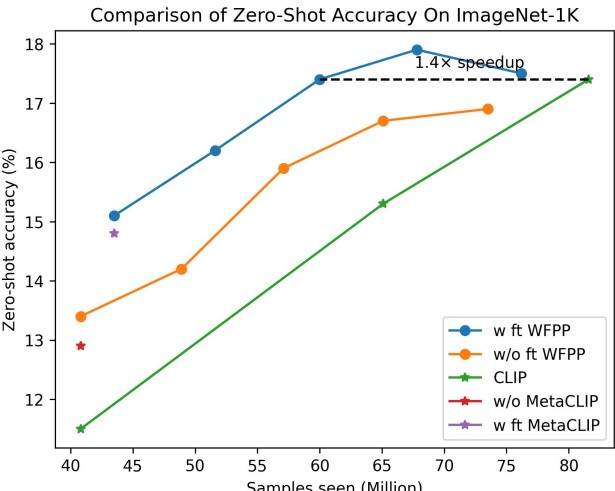

Figure 3: **Zero-shot accuracy on ImageNet-1K classification.** The CC3M dataset (Sharma et al., 2018) was pruned with WFPP. We see that CLIP trained on the WFPP-pruned data achieved comparable performance with CLIP trained on unpruned data, but uses only approximately 70% of original training data (indicated by the 1.4x speed-up mark). On the left, we see examples of the performance of data pruned with the MetaCLIP method, which remains below the performance of data pruned with WFPP. The image encoder is ViT-B-16 (Dosovitskiy et al., 2021). The "ft" is an initial for fine-tuning. "Samples seen" refers to the number of samples processed during pre-training.

# A  APPENDIX

## A.1  MORE EVALUATION

For comparison with MetaCLIP, we pre-trained our model by pruning 50% of the data in CC3M using the same metadata as MetaCLIP, with the same pre-training and fine-tuning settings as WFPP. As shown in Figure 3, WFPP pre-trained on CC3M only requires 70% of the sample size required by CLIP.

**Zero-shot Robustness Evaluation:** The performance of the pre-trained model on the CC3M (Sharma et al., 2018) dataset is consistent with the pre-trained model on the CC12M dataset, as seen in Table 13. When using an 80% subset of CC3M for pre-training, the model achieves the best average performance on these datasets. Additionally, when comparing our method with MetaCLIP, our method outperforms MetaCLIP on all datasets and exceeds MetaCLIP by an average of 0.29% in zero-shot robustness evaluation.

Table 13: **Zero-shot robustness evaluation:** Comparison of zero-shot accuracy performance between CLIP trained on the original data and CLIP trained on data pruned with the MetaCLIP method and with WFPP, on various datasets. The image encoder is ViT-B-16, and the pre-trained dataset is CC3M (Sharma et al., 2018). The model is fine-tuned for another epoch on the entire dataset.

| Dataset | CLIP | MetaCLIP 50% | WFPP | | | | |
|---|---|---|---|---|---|---|---|
| | | | 50% | 60% | 70% | 80% | 90% |
| ImageNet-A | 4.09 | 3.20 | 3.52 | 4.04 | 3.84 | **4.49** | 3.93 |
| ImageNet-O | 21.60 | 18.90 | 19.20 | 20.80 | 21.60 | **23.10** | 21.70 |
| ImageNet-R | **20.72** | 15.55 | 16.31 | 17.48 | 19.28 | 20.63 | 20.26 |
| ImageNet Sketch | 8.09 | 5.28 | 5.54 | 6.43 | 7.46 | 8.19 | **8.65** |
| ImageNetV2 | 14.74 | 12.79 | 13.02 | 14.04 | 14.63 | **15.28** | 14.77 |
| ObjectNet | 10.15 | 8.25 | 9.12 | 9.34 | 10.09 | 10.67 | **10.95** |
| Average | 13.20 | 10.83 | 11.12 | 12.02 | 12.82 | **13.73** | 13.38 |

Table 14: **Zero-shot accuracy on more classification datasets.** The image encoder is ViT-B-16, and the pre-training dataset is CC3M (Sharma et al., 2018). The model is fine-tuned for another epoch on the entire dataset.

| Dataset | CLIP | MetaCLIP 50% | WFPP 50% | 60% | 70% | 80% | 90% |
|---|---|---|---|---|---|---|---|
| Food-101 | 11.36 | 10.70 | 11.36 | 11.36 | 11.39 | 12.00 | 10.28 |
| CIFAR-10 | 43.07 | 44.72 | 39.60 | 39.95 | 41.49 | 41.63 | 39.47 |
| CIFAR-100 | 18.18 | 17.29 | 15.56 | 17.53 | 19.51 | 20.53 | 16.30 |
| CUB200 | 3.30 | 3.00 | 3.26 | 3.12 | 2.95 | 3.50 | 3.78 |
| SUN397 | 33.30 | 28.30 | 28.38 | 30.43 | 33.12 | 34.50 | 33.46 |
| Cars | 0.88 | 0.88 | 0.72 | 0.87 | 1.02 | 0.62 | 0.91 |
| Aircraft | 0.75 | 1.02 | 1.54 | 1.33 | 1.38 | 1.02 | 1.33 |
| DTD | 10.32 | 10.64 | 11.81 | 13.40 | 13.09 | 11.06 | 12.13 |
| OxfordPets | 11.90 | 9.26 | 12.56 | 10.04 | 13.94 | 14.25 | 12.08 |
| Caltech-101 | 46.40 | 39.82 | 40.89 | 42.15 | 46.46 | 46.22 | 46.73 |
| Kinetics700 | 13.12 | 11.43 | 11.89 | 12.72 | 13.52 | 13.40 | 13.18 |
| Flowers102 | 1.90 | 1.22 | 2.57 | 1.63 | 1.66 | 1.64 | 1.89 |
| MNIST | 10.10 | 10.09 | 8.92 | 9.54 | 12.58 | 7.77 | 13.26 |
| STL10 | 80.51 | 72.69 | 76.48 | 76.20 | 79.16 | 82.09 | 81.03 |
| EuroSAT | 16.10 | 11.68 | 18.64 | 14.02 | 20.00 | 7.70 | 18.56 |
| Resisc45 | 20.02 | 16.52 | 17.83 | 16.70 | 18.70 | 20.52 | 20.30 |
| GTSRB | 8.65 | 4.77 | 5.87 | 4.24 | 8.35 | 6.93 | 6.29 |
| KITTI | 34.63 | 39.10 | 22.39 | 28.88 | 40.64 | 29.34 | 28.68 |
| Country211 | 0.69 | 0.62 | 0.60 | 0.57 | 0.64 | 0.80 | 0.61 |
| PCAM | 56.14 | 50.05 | 50.05 | 60.18 | 50.00 | 50.01 | 56.30 |
| UCF101 | 25.09 | 21.25 | 20.20 | 21.78 | 23.02 | 25.40 | 24.21 |
| CLEVR | 12.09 | 19.93 | 13.22 | 11.60 | 11.90 | 13.39 | 9.57 |
| HatefulMemes | 50.94 | 52.97 | 49.61 | 52.71 | 55.95 | 54.38 | 50.74 |
| SST2 | 50.08 | 48.76 | 50.08 | 50.08 | 49.48 | 49.26 | 49.92 |
| ImageNet | 17.36 | 14.78 | 15.16 | 16.23 | 17.37 | 17.91 | 17.50 |
| **Average** | 23.08 | 16.72 | 21.18 | 23.49 | **23.49** | 22.63 | 22.74 |

Table 15: **Zero-shot Image-Text Retrieval:** We evaluate CLIP trained on the original data vs. CLIP trained on data pruned with the MetaCLIP methods and with WFPP in terms of image-text retrieval performance on both COCO and Flickr30k datasets. The image encoder is ViT-B-16, and the pre-trained dataset is CC3M (Sharma et al., 2018). The model is fine-tuned for another epoch on the entire dataset.

| Model | Sample Size | Text Retrieval | | | | | | Image Retrieval | | | | | |
|---|---|---|---|---|---|---|---|---|---|---|---|---|---|
| | | Flickr30k | | | COCO | | | Flickr30k | | | COCO | | |
| | | R@1 | R@5 | R@10 | R@1 | R@5 | R@10 | R@1 | R@5 | R@10 | R@1 | R@5 | R@10 |
| CLIP | 100% | 24.79 | 48.30 | 58.48 | 11.91 | 29.56 | 40.20 | 31.56 | 60.16 | 70.22 | 15.98 | 37.52 | 49.06 |
| MetaCLIP | 50% | 17.02 | 37.02 | 48.11 | 12.42 | 30.14 | 40.58 | 23.67 | 46.94 | 58.19 | 8.90 | 23.05 | 32.87 |
| WFPP | 50% | 17.24 | 37.04 | 47.04 | 8.38 | 23.09 | 32.85 | 23.18 | 45.27 | 58.68 | 11.28 | 29.08 | 39.90 |
| | 60% | 19.59 | 41.22 | 52.01 | 9.86 | 25.90 | 36.03 | 25.74 | 50.49 | 61.64 | 14.16 | 31.96 | 43.18 |
| | 70% | 22.07 | 46.27 | 56.79 | 11.02 | 28.10 | 38.60 | 29.59 | 56.11 | 67.85 | 15.28 | 34.92 | 45.86 |
| | 80% | 24.20 | 47.02 | 57.53 | 11.95 | 29.75 | 40.53 | 31.07 | 57.40 | 66.77 | 16.34 | 37.24 | 48.52 |
| | 90% | 25.50 | 48.60 | 59.59 | 12.36 | 30.64 | 41.39 | 30.18 | 61.54 | 70.61 | 16.14 | 37.46 | 49.40 |

**Zero-shot Classification on More Datasets** The zero-shot classification performance across 25 datasets is consistent with the results from the model pre-trained on the CC3M dataset, as shown in Table 14. CLIP trained on data pruned with WFPP achieves the best average performance when using a 70% subset, outperforming CLIP pre-trained on the entire dataset by 0.41%. Additionally, compared to CLIP trained on data pruned with the MetaCLIP method, our method substantially exceeds its average performance by 4.46%.

**Zero-shot Retrieval** As shown in Table 15, models pre-trained on the CC3M dataset demonstrate that CLIP trained on data pruned with WFPP has a substantial advantage in zero-shot image-text retrieval tasks compared to CLIP trained on the original data and CLIP trained on data pruned with the MetaCLIP method.

Table 16: **Zero-shot classification accuracy on ImageNet-1K.** We pre-trained models by sampling image-text pairs sorted according to Equation 3, with sampling rates of 50% based on different thresholds $t$, specifically 1e-6, 1e-7, and 1e-8. The pre-training dataset is **CC12M** (Changpinyo et al., 2021), and the image encoder used is ViT-B-16 (Dosovitskiy et al., 2021). "Samples seen" refers to the proportion of the dataset processed during pre-training, with 100% set as 1.00. w/ft and w/o/ft is with and without fine-tuning.

| Method | Sample Size | threshold | w/o/ft | w/ft | Samples seen (w/ft) |
|--------|-------------|-----------|--------|------|---------------------|
| CLIP | 4.65M (50%) | ✗ | 28.2 | 30.2 | 0.53× |
| | 4.65M (50%) | 1e-6 | 29.0 | 30.7 | 0.53× |
| WFPP | 4.65M (50%) | 1e-7 | 29.8 | 31.3 | 0.53× |
| | 4.65M (50%) | 1e-8 | 29.7 | 31.2 | 0.53× |

Table 17: **Zero-shot classification accuracy on ImageNet-1K.** We use BLIP Li et al. (2022a) to generate **synthetic captions** for the CC3M dataset. We then prune 50% of these synthetic captions using random and WFPP methods. The model is pre-trained for 30 epochs and fine-tuned on the original dataset for 1 epoch. The image encoder employed is ViT-B-16 Dosovitskiy et al. (2021). The threshold value $t$ in Equation 2 is set to $10^{-7}$.

| Method | data | Sample Size | w/o/ft | w/ft | Samples seen (w/ft) |
|--------|------|-------------|--------|------|---------------------|
| CLIP | synthetic captions | 50% | 7.7 | **12.6** | 0.53× |
| WFPP | | 50% | 7.8 | **13.0** | 0.53× |

## A.2 THRESHOLD SELECTION FOR WFPP

Based on the idea that high-frequency and low-frequency words should be treated differently, we select the threshold using Equation 2. We aim to decrease the sampling probability of high-frequency words while retaining low-frequency words. For words with frequency $f(w_i)$ less than $t$ (i.e., very rare words), we set $P(w_i)$ to 1. These very rare words do not affect the probability of the text being discarded. Specifically, for the cc12m dataset, setting $t$ to 1e-6 means words with count less than 206 are assigned $P(w_i) = 1$; setting $t$ to 1e-7 corresponds to words with a count less than 20 being assigned $P(w_i) = 1$; when $t$ is set to 1e-8, all words are considered by Equation 2 (since no words have frequency less than $t$). We also provide zero-shot classification accuracy with three different thresholds: 1e-6, 1e-7, and 1e-8, As shown in Table 16. When we set the threshold $t$ to 1e-6, the zero-shot classification accuracy is lower than when $t$ is 1e-7 or 1e-8, but it still outperforms the random pruning method. The performances for thresholds $1 \times 10^{-7}$ and $1 \times 10^{-8}$ are similar.

## A.3 SYNTHETIC CAPTIONS

Some works utilize synthetic image captions to pre-train CLIP Vasu et al. (2024); Yang et al. (2023). While these synthetic captions are of higher quality than web captions, they do not ensure a balanced word distribution. To address this imbalance, WFPP can be applied to the synthetic captions datasets. As shown in Table 17, we generated synthetic captions for the CC3M dataset using BLIP Li et al. (2022a). We then pruned 50% of the synthetic data using random pruning and WFPP to pre-train CLIP. Our results show that WFPP outperforms the random pruning method by 0.4% in zero-shot classification accuracy on the zero-shot ImageNet-1k classification.

## A.4 WORD AND TOKEN FREQUENCY

We also evaluate the model's performance through token frequency data pruning. As indicated in Table 18, the performance based on word and token frequency is similar. This is because most words correspond to individual tokens. Although some words are tokenized into two or more tokens, this has a limited impact on overall performance.

## A.5 DATA ANALYSIS

We analyzed the data categories before and after applying WFPP. According to Table 19, there is not a significant change in the percentage of word categories. However, when this data is combined with

Table 18: **Zero-shot classification accuracy on ImageNet-1K.** We pre-train models by sampling image-text pairs sorted according to Equation 3 at sampling rates 50% based on **word frequency** and **token frequency**. The pre-training dataset is **CC12M** (Changpinyo et al., 2021), and the image encoder used is ViT-B-16 (Dosovitskiy et al., 2021).

| Method | Sample Size | tokenizer | w/o/ft | w/ft | Samples seen (w/ft) |
|---|---|---|---|---|---|
| WFPP | 4.65M (50%) | word | 29.8 | 31.3 | 0.53× |
| | 4.65M (50%) | token | 29.8 | 31.5 | 0.53× |

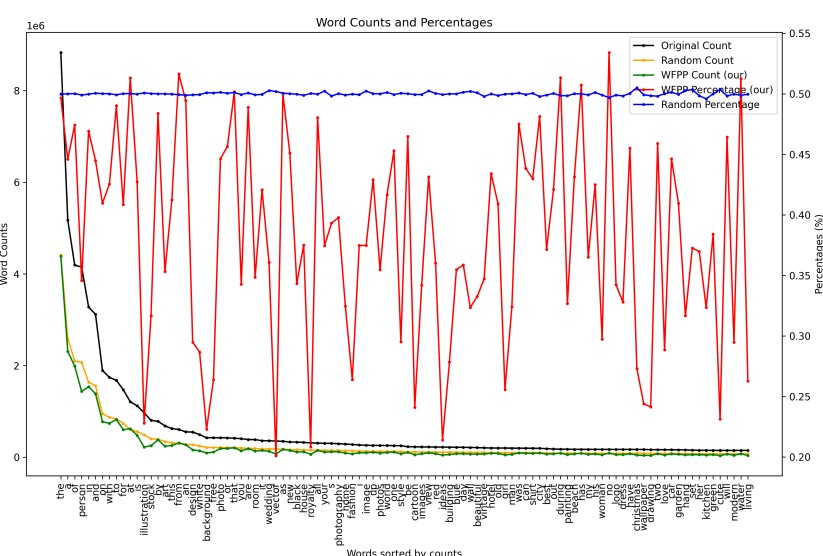

Figure 4: **Word Distribution:** The top-100 words in CC12M (Changpinyo et al., 2021) are shown after pruning 50% of image-text pairs using Random (orange) and WFPP (green) methods, and before pruning (black). We then calculate the word percentages for Random and WFPP before and after pruning. Words are ordered by frequency before pruning. The left Y-axis is the number of words and the left Y-axis is the percentage of words which is the number of words before data pruning divided by the number of words after data pruning.

the analysis presented in Figure 4, it becomes apparent that WFPP prunes more high-frequency noun words while retaining more low-frequency nouns. WFPP improves word distribution by selectively pruning high-frequency words, thereby achieving a more balanced distribution. In addition, we visualize the word distribution for the first and second part of the data after applying WFPP, as shown in Figure 5. The second part has many more high-frequency words than the first part and the high-frequency words percentage of the second part is substantially higher than 50%. Moreover, we also analyzed human-labeled datasets by applying WFPP, such as COCO Lin et al. (2014) and VG Krishna et al. (2017). As illustrated in Figure 6 and Figure 7, the patterns observed in these figures are similar to those in the CC12M dataset. By applying the WFPP method, more high-frequency words are pruned from the first part of the dataset, while the second part retains a higher number of these words.

Table 19: We analyzed the percentages and counts resulting from random and frequency-based pruning methods. Using the CC12M dataset, we pruned 50% of the original texts. NN means noun, JJ means adjective, VB means verb.

| Method | NN | | JJ | | VB | | OTHER | | Total |
|---|---|---|---|---|---|---|---|---|---|
| Before sampling | 103,469,117 | (50.34%) | 10,245,828 | (4.98%) | 10,649,943 | (5.18%) | 81,351,966 | (39.61%) | 205,716,854 |
| Random | 51,648,996 | (50.25%) | 5,120,215 | (4.98%) | 5,319,414 | (5.18%) | 40,666,145 | (39.59%) | 102,754,770 |
| WPFF | 47,149,193 | (50.48%) | 4,491,732 | (4.81%) | 5,153,531 | (5.52%) | 36,596,727 | (39.19%) | 93,391,183 |

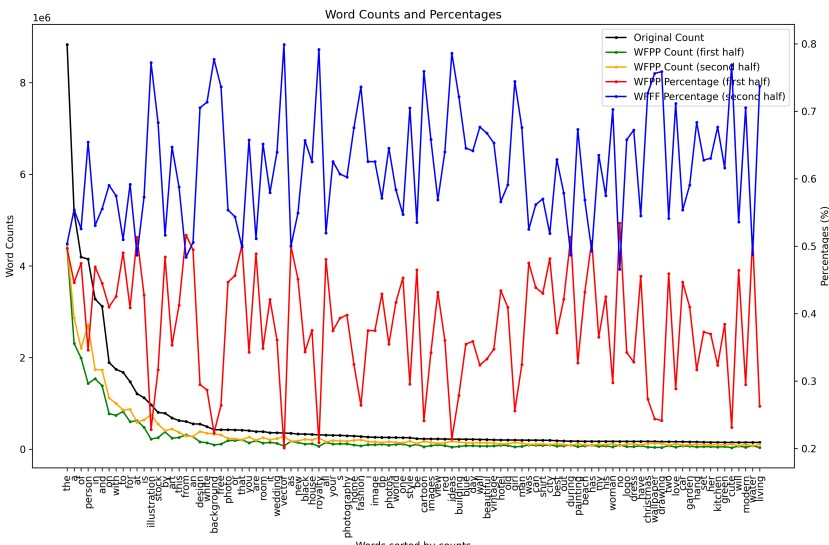

Figure 5: **Word Distribution:** The top 100 words from CC12M (Changpinyo et al., 2021) are presented in two parts: the first and second 50% of the image-text pairs using the WFPP method. We then calculate the word percentages for WFPP before and after pruning. Words are ordered by frequency before pruning. The left Y-axis is the number of words and the left Y-axis is the percentage of words which is the number of words before data pruning divided by the number of words after data pruning.

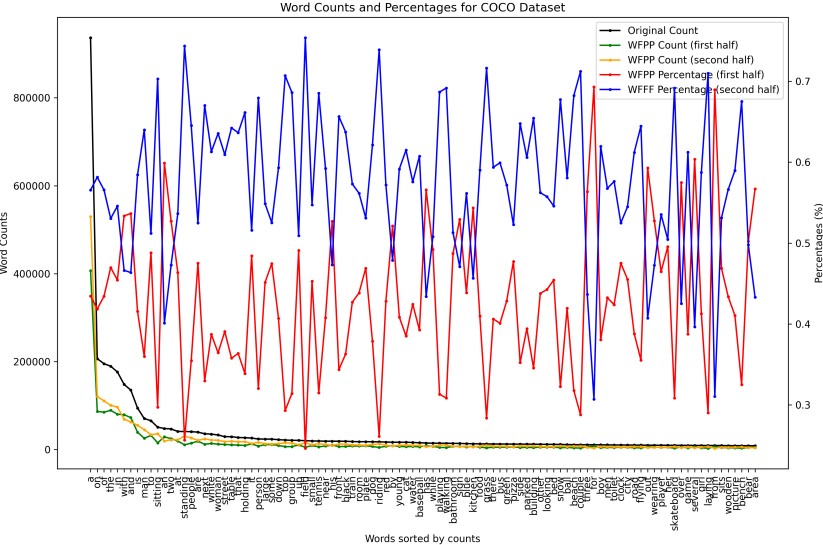

Figure 6: **COCO Dataset Word Distribution:** The top 100 words from COCO (Lin et al., 2014) dataset are presented in two parts: the first and second 50% of the image-text pairs using the WFPP method. We then calculate the word percentages for WFPP before and after pruning. Words are ordered by frequency before pruning. The left Y-axis is the number of words and the left Y-axis is the percentage of words which is the number of words before data pruning divided by the number of words after data pruning.

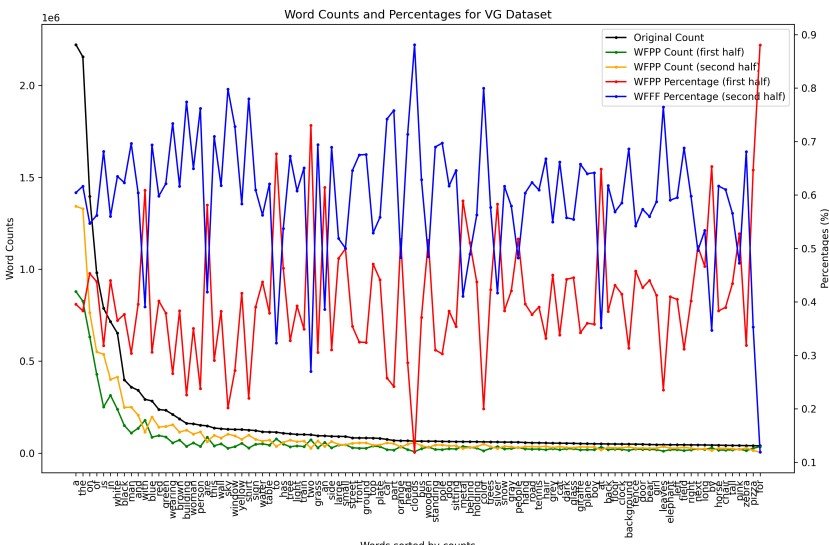

Figure 7: **VG Dataset Word Distribution:** The top 100 words from VG (Krishna et al., 2017) dataset are presented in two parts: the first and second 50% of the image-text pairs using the WFPP method. We then calculate the word percentages for WFPP before and after pruning. Words are ordered by frequency before pruning. The left Y-axis is the number of words and the left Y-axis is the percentage of words which is the number of words before data pruning divided by the number of words after data pruning.

# B   DATA EXAMPLES

We randomly select example words from the dictionary and list them with their corresponding $P(w_i)$ values. As shown in Table 20, frequent words have a higher probability of being discarded than infrequent words.

Table 20: Randomly selected words and their $P(w_i)$ values.

| Word | $P(w_i)$ | Word | $P(w_i)$ | Word | $P(w_i)$ |
|---|---|---|---|---|---|
| , | 0.9996 | the | 0.9995 | person | 0.9993 |
| in | 0.9992 | > | 0.9991 | it | 0.9976 |
| dream | 0.9792 | latest | 0.9736 | material | 0.9572 |
| prepared | 0.9305 | brighten | 0.9015 | macau | 0.8777 |
| lesser | 0.8467 | billionaire | 0.8321 | lawns | 0.8252 |
| authenticity | 0.8249 | raging | 0.8090 | apricots | 0.7670 |
| fidget | 0.7309 | decathlon | 0.7240 | quadratic | 0.7084 |
| natura | 0.7078 | momo | 0.6963 | squaw | 0.6619 |
| pats | 0.6511 | futurist | 0.6425 | ardmore | 0.6220 |
| knott | 0.5615 | dungarees | 0.5464 | hairdryer | 0.5347 |
| ankole | 0.5051 | escolta | 0.4991 | incubation | 0.4831 |
| kalibo | 0.4240 | okefenokee | 0.4240 | lusitano | 0.3710 |
| whe | 0.3649 | hubcap | 0.3384 | compromises | 0.3239 |
| bydgoszcz | 0.3239 | ews | 0.3162 | zar | 0.2737 |
| actuators | 0.2333 | skeet | 0.2333 | tawang | 0.2333 |
| dregs | 0.2222 | urad | 0.1982 | alternated | 0.1982 |
| essequibo | 0.1854 | ilent | 0.1719 | cline | 0.1719 |
| holtz | 0.0929 | sacramental | 0.0742 | barish | 0.0742 |
| stippled | 0.0742 | junina | 0.0543 | rafted | 0.0103 |

Additionally, we randomly selected texts from the CC12M dataset and calculated their corresponding $S(w_i)$ values. The results are presented in Table 21.

Table 21: Randomly selected example texts and their corresponding $S(w_i)$. The texts are select from CC12M (Changpinyo et al., 2021).

| **Texts** | $S(w_i)$ |
|---|---|
| Girona Beach with <PERSON> | 0.06019 |
| Black Canyon of the Gunnison | 0.05801 |
| The Sutton Condominium Residents Lounge Couch | 0.05748 |
| Sagas of the Gray Seas: Sleipnir's Hoof | 0.05624 |
| <PERSON>- The Boiler Room | 0.02104 |
| <PERSON>deep in a drift | 0.02041 |
| New chef at the Portsea Hotel, <PERSON> | 0.02033 |
| A bed or beds in a room at <PERSON>'s Guest House and Tours | 0.00892 |
| Creative Design For Blue Red And An Orange Wallpaper Photo | 0.00882 |
| The Lord of the Rings Game HD Wallpaper 1920x1080 | 0.00861 |
| A small dog stock photography | 0.00792 |
| <PERSON>family with a child | 0.00662 |

