# OpenReview forum: "Enhancing Vision-Language Model Pre-training with Image-text Pair Pruning Based on Word Frequency"
_ICLR.cc/2025/Conference — Submitted to ICLR 2025_

### Official Review · Reviewer_zmLE · 2024-10-30

**Soundness:** 3
**Presentation:** 2
**Contribution:** 2
**Rating:** 3
**Confidence:** 4

**Summary:**

This paper introduces a data pruning method for CLIP that focuses on removing frequently occurring data samples. The experimental results show that the pruned model outperforms the original CLIP across several datasets and downstream tasks, indicating that eliminating high-frequency samples can enhance CLIP's training efficiency and performance.

**Strengths:**

Simplicity: The proposed approach is intuitive and easy to implement, which enhances its potential for broader application.

Performance Gains: The method consistently outperforms the original CLIP across most datasets, demonstrating that the removal of high-frequency samples leads to improved model performance.

**Weaknesses:**

Insufficient Coverage of Related Work: The paper lacks citations and discussions of key related works in data pruning. Crucial references such as [1-3] are missing. Including these would better position the contribution within the current research landscape.

Limited Baseline Comparisons: The experimental comparison is restricted to the CLIP and MetaCLIP, neglecting other state-of-the-art (SOTA) data pruning methods. A broader comparison with cutting-edge approaches would strengthen the evaluation and provide a more comprehensive assessment of the method's effectiveness.

Narrow Model Scope: The pruning method is evaluated only on CLIP, without experiments on other architectures such as BLIP or Llava, which are also highly relevant for vision-language and generative tasks. Additionally, the paper does not explore how the proposed method performs on more challenging vision-language tasks like VQA or image captioning. Expanding the experimental scope to include both different models and more challenging tasks would help to establish the method’s generalizability and effectiveness across diverse architectures and applications.

[1] Beyond Neural Scaling Laws: Beating Power Law Scaling via Data Pruning.

[2] When Less is More: Investigating Data Pruning for Pretraining LLMs at Scale.

[3] Data Pruning via Moving-one-Sample-out.

**Questions:**

Please see weakness.

---

> ### Author Response · Authors · 2024-11-25
>
> Dear Reviewer,
>
> Thank you for your thorough review of our submission. For your questions, please check our overall answer and the revised paper to get the new results and analyses.

---

> > ### Comment · Reviewer_zmLE · 2024-11-30
> >
> > As most of my concerns remain unaddressed, except for the addition of related work on data pruning, I lower my rating to 3.

---

> > > ### Author Response · Authors · 2024-12-03
> > >
> > > Dear Reviewer,
> > >
> > > We appreciate your taking the time to review our paper. We point out that our approach is computationally light-weight compared to many other approaches in the literature. Also, it has the potential to complement other approaches that do not leverage word-frequency information.

---

### Official Review · Reviewer_eLe2 · 2024-11-03

**Soundness:** 2
**Presentation:** 3
**Contribution:** 2
**Rating:** 8
**Confidence:** 3

**Summary:**

This paper introduces Word-Frequency-based Image-Text Pair Pruning (WFPP), a novel approach designed to enhance the selection of image-text pairs for more effective CLIP pre-training.

**Strengths:**

1. The paper is clearly written and structured in a way that is easy to understand.

2. The experiments conducted on robustness classification tasks are thoroughly executed.

**Weaknesses:**

1. While CLIP is over three years old and attention has shifted to newer multimodal large language models (MLLMs), the impact of WFPP on larger models such as LLaVA remains under-explored.
2. MetaCLIP [1] has demonstrated impressive success in establishing a comprehensive pipeline for entire CLIP pre-training. However, the WFPP experiments are conducted using CC3M and CC12M datasets, which are significantly smaller than the 400M images used in the original CLIP training. This raises questions about the scalability and usefulness of WFPP when applied to larger datasets.
3. Although the authors claim that WFPP can enhance performance across a broad range of downstream tasks, the comparison baselines only include the original CLIP, which may not sufficiently demonstrate the method's relative effectiveness.
4. The proposed method employs techniques that are simple and already widely used in the NLP community. Therefore, more comprehensive experiments and detailed discussions are crucial to substantiate its uniqueness and effectiveness.

[1] Demystifying clip data. ICLR 2024

**Questions:**

1. Is there any analysis or discussion in the paper that supports the effectiveness of WFPP when applied to much larger datasets, such as LAION?

2. It would be beneficial for the authors to compare WFPP with improved CLIP pre-training baselines, such as DeCLIP [2], FILIP [3], and UniCLIP [4], to better showcase the strengths of their approach.

3. Given that other works, such as TinyCLIP [5] and MoPE-CLIP [6], have achieved competitive performance by pruning and training CLIP models on datasets like CC12M or YFCC15M, a discussion and comparison with these studies would provide a more comprehensive view of WFPP's comparative advantages.

4. Numerous data processing techniques have been proposed to enhance CLIP pre-training, such as DataCompDR in MobileCLIP [7] and the OFA module in ALIP [8]. How does WFPP compare to these strategies in terms of improving pre-training performance?

[2] Supervision exists everywhere: A data-efficient contrastive language-image pre-training paradigm. ICLR 2022

[3] Filip: fine-grained interactive language-image pre-training. ICLR 2022

[4]  Uniclip: Unified framework for contrastive language-image pretraining. NeurIPS 2022

[5] Tinyclip: Clip distillation via affinity mimicking and weight inheritance. ICCV 2023

[6] Mope-clip: Structured pruning for efficient vision-language models with module-wise pruning error metric. CVPR 2024

[7] Mobileclip: Fast image-text models through multi-modal reinforced training. CVPR 2024

[8] Alip: Adaptive language-image pre-training with synthetic caption. ICCV 2023

---

> ### Author Response · Authors · 2024-11-25
>
> Dear Reviewer,
>
> Thank you for your thorough review of our submission. Please also check our overall answer. Here are our answers to your specific questions:
>
> Question 1: Is there any analysis or discussion in the paper that supports the effectiveness of WFPP when applied to much larger datasets, such as LAION?
>
> Due to limited computing resources, we are unable to pre-train our models on the entire LAION-400M dataset. This limitation is discussed in Section 5 (Conclusions and Outlook).
>
> Questions 2 and 3: It would be beneficial for the authors to compare WFPP with improved CLIP pre-training baselines, such as DeCLIP [2], FILIP [3], and UniCLIP [4], to better showcase the strengths of their approach. Given that other works, such as TinyCLIP [5] and MoPE-CLIP [6], have achieved competitive performance by pruning and training CLIP models on datasets like CC12M or YFCC15M, a discussion and comparison with these studies would provide a more comprehensive view of WFPP's comparative advantages.
>
> Given that other works, such as TinyCLIP [5] and MoPE-CLIP [6], have achieved competitive performance by pruning and training CLIP models on datasets like CC12M or YFCC15M, a discussion and comparison with these studies would provide a more comprehensive view of WFPP's comparative advantages.
>
> In the revised paper, we added a discussion of these approaches in the related work. WPFF is a data pruning method and the other methods mentioned by the reviewer use other approaches to reduce computation, it is possible to add WPFF on top of any of the other methods to achieve additional gains.
>
> Question 4: Numerous data processing techniques have been proposed to enhance CLIP pre-training, such as DataCompDR in MobileCLIP [7] and the OFA module in ALIP [8]. How does WFPP compare to these strategies in terms of improving pre-training performance?
>
> In the revised paper, we carried out a new experiment to evaluate our method on synthetic captions (Table 17). MobileCLIP[7] and Alip[8] generate synthetic captions through pre-training on large-scale datasets. MobileCLIP uses CoCa[9] to create multiple synthetic captions, while Alip employs OFA_base[10] for caption generation. While these synthetic captions offer higher quality than web captions, they still lack balanced word distribution. WFPP could enhance these methods by addressing this distribution imbalance. Therefore, we applied WFPP to synthetic captions generated by BLIP for CLIP pre-training. As shown in Table 17, WFPP surpassed the random pruning method by 0.4% in zero-shot classification accuracy on ImageNet-1k.
>
> In summary, WFPP is a data-pruning method based on word frequency. It can be applied not only to prune pre-trained datasets but also to synthetic captions. It is essential to include these applications in our related work section and present the results of applying WFPP to some of these models.
>
> [1]  Demystifying clip data. ICLR 2024
>
> [2] Supervision exists everywhere: A data-efficient contrastive language-image pre-training paradigm. ICLR 2022
>
> [3] Filip: fine-grained interactive language-image pre-training. ICLR 2022
>
> [4] Uniclip: Unified framework for contrastive language-image pretraining. NeurIPS 2022
>
> [5] Tinyclip: Clip distillation via affinity mimicking and weight inheritance. ICCV 2023
>
> [6] Mope-clip: Structured pruning for efficient vision-language models with module-wise pruning error metric. CVPR 2024
>
> [7] Mobileclip: Fast image-text models through multi-modal reinforced training. CVPR 2024
>
> [8] Alip: Adaptive language-image pre-training with synthetic caption. ICCV 2023
>
> [9] CoCa: Contrastive Captioners are Image-Text Foundation Models. Transactions on Machine Learning Research 2022
>
> [10] Ofa: Unifying architectures, tasks, and modalities through a simple sequence-to-sequence learning framework. ICML, 2022.

---

> ### Comment · Reviewer_eLe2 · 2024-11-26
> **Response to Rebuttal**
>
> Thank you for responding to my questions. The response addresses my concerns well, and I will raise my score to 6, recommending acceptance.
>
> I suggest the authors include the discussion on Efficient Language-Image Pre-training Methods and additional experiments in the final version.
>
> By the way, conducting an experiment that combines your WFPP with one of these efficient methods on a smaller dataset, such as CC3M, could further strengthen the soundness of this work. As the discussion period has been extended, if you are able to provide this additional experiment, I would be happy to reconsider my score and potentially raise it further.

---

> > ### Author Response · Authors · 2024-11-30
> >
> > Dear Reviewer,
> >
> > Thank you for your positive feedback and for raising the score.
> >
> > We appreciate your suggestion about combining WFPP with efficient methods. We carried out a new experiment to evaluate our method on DeCLIP. We pre-trained DeCLIP on the CC3M dataset  with both random pruning and WFPP, in each case, moving 50% of image-text pairs. We trained for 32,000 steps on 8 A5000 GPUs. We use the same code and settings as DeCLIP. This was the most informative experiment that we could carry out, given the time constraints. Here are the results:
> >
> > **Zero-shot classification accuracy on ImageNet-1K.**  DeCLIP pre-trained on the whole CC3M dataset (32000 steps) and 50% of the whole data set, with both random pruning and WFPP (28000 steps with the pruned data followed by 4000 steps of fine-tuning on the whole data set).
> >
> > The image encoder employed is ViT-B-32 . The threshold value $t$ in Equation (1) is set to $10^{-7}$.
> > | Model   | Method | Sample Size | w/o ft | w/ ft    | Samples Seen (w/ft) |
> > |---------|--------|-------------|--------|----------|---------------------|
> > | DeCLIP  | ✗      | 100%        | 14.1   | ✗        | 26.6M               |
> > | DeCLIP        | Random | 50%         | 2.2    | 3.0      | 26.6M               |
> > | DeCLIP       | WFPP   | 50%         | 13.2   | **13.8** | 26.6M               |
> >
> > When applying our WFPP approach to DeCLIP, the performance approaches that of the model pre-trained on the entire CC3M data set. We were surprised at how harmful random pruning turned out to be in contrast to WFPP. We believe that WFPP outperforms random pruning because leaves more diverse samples for the Nearest-Neighbor Supervision strategy used by DeCLIP. These results demonstrate the value of WFPP added on top of DeCLIP.

---

> > > ### Comment · Reviewer_eLe2 · 2024-12-03
> > >
> > > Thank you for your additional experiments. I will raise my score. I recommend including these discussions and experiments in the next version of your paper. By the way, my only remaining concern is the effectiveness of WFPP on large-scale datasets, which are used for CLIP or OpenCLIP training.

---

> > > > ### Author Response · Authors · 2024-12-03
> > > >
> > > > Dear Reviewer,
> > > >
> > > > Thank you. We will take your suggestions on board.

---

### Official Review · Reviewer_VHZe · 2024-11-04

**Soundness:** 3
**Presentation:** 3
**Contribution:** 3
**Rating:** 8
**Confidence:** 3

**Summary:**

This paper hypothesizes that a balanced word-frequency distribution in a dataset can improve the training efficiency of Visual Language Models (VLMs). Through extensive experiments, the authors demonstrate that their proposed method, Word-Frequency-based Image-Text Pair Pruning (WFPP), achieves better performance with a reduced training budget.

**Strengths:**

1. The approach is straightforward and constructive.
2. The authors conduct comprehensive experiments to validate WFPP’s effectiveness.
3. The analysis on text length normalization provides valuable insights.
4. The authors make all code and data publicly available, enhancing reproducibility.

**Weaknesses:**

1. The paper lacks theoretical analysis explaining why pruning based on word frequency is effective for image-text alignment.

**Questions:**

1. What does “w/o ft WFPP” mean in Figure 1?
2. Existing methods emphasize the importance of re-captioning. Would the proposed method be affected by re-captioning? What if, instead of deleting image-text pairs, we re-captioned the images?
3. WFPP uses word frequency as an indicator. Have the authors considered using alternative criteria, such as n-grams or token frequency?

---

> ### Author Response · Authors · 2024-11-25
>
> Dear Reviewer,
>
> Thank you for your thorough review of our submission. Please also check our overall answer. Here are our answers to your specific questions:
>
> Question 1: What does “w/o ft WFPP” mean in Figure 1?
>
> w/o ft WFPP means with out fine-tuning. We have added it to the caption of Fig. 1 for clarity.
>
> Question 2: Existing methods emphasize the importance of re-captioning. Would the proposed method be affected by re-captioning? What if, instead of deleting image-text pairs, we re-captioned the images?
>
> We carried out an extra experiment and added a table (Table 17) to the revised version of the paper to answer this question. We generated synthetic captions for the CC3M dataset using  BLIP [1] and demonstrated that WFPP delivers a performance boost on top of the use of synthetic captions. This result suggests that synthetic captions could not replace WFPP, but rather that both could be used together.
>
> Question 3: WFPP uses word frequency as an indicator. Have the authors considered using alternative criteria, such as n-grams or token frequency?
>
> We carried out a new experiment using token frequency and added it to the revised paper (Table 18). The result was comparable to using word frequency. N-gram selection would be interesting to explore in future work, however, since n-grams are less frequent than words, we don’t expect the frequency unbalance issue to be as great of a problem.
>
> [1] BLIP: Bootstrapping Language-Image Pre-training for Unified Vision-Language Understanding and Generation.

---

### Official Review · Reviewer_TRgD · 2024-11-04

**Soundness:** 3
**Presentation:** 3
**Contribution:** 2
**Rating:** 3
**Confidence:** 3

**Summary:**

This paper introduces Word-Frequency-based Image-Text Pair Pruning (WFPP), a method to enhance the pretraining of Vision-Language Models (VLMs) by pruning the training dataset based on word frequency. The authors argue that reducing the dominance of high-frequency words leads to a better-balanced word-frequency distribution, which in turn improves the training of word embedding models. The experiments show that their approach, when applied to training a CLIP model, improves performance on various downstream tasks and speeds up pre-training by using fewer samples.

**Strengths:**

- The proposed method is simple yet effective, and the authors demonstrate its superiority on various benchmarks over other pruning methods for both pre-training and fine-tuning.

- The paper is well-written, and the figures and tables are clear and support the narrative effectively.

**Weaknesses:**

- The proposed method is verified on small scale dataset (CC3M and CC12M) with a large gap of accuracy against existsing methods (e.g., CLIP, MetaCLIP.) pretrained on larger scales (i.e., CommonCrawl 400M). For instance, the zero-shot classification accuracy on ImageNet 1K is around 35%+ for WFPP, but CLIP and MetaCLIP enjoys around 70%. Thus it may still remains unsure if WFPP can be extent to larger scales, expecially as a method to prune the large scale pre-training dataset.

- The probability to discard a sample decreases with the increasing length of input text, and this holds for both Equation 3 and Equation 4. This is intuitively unreasonable since longer input would be less likely to be discarded. The text length is set to 32 by default, but in practice the text lenght can vary a lot and more experiments w.r.t. the text length should be discussed.

**Questions:**

- The paper uses a straightforward approach to calculate word frequency, which is essential for the WFPP method. It would be beneficial if the authors could discuss the choice of this particular frequency calculation over other potential methods, such as term frequency-inverse document frequency (TF-IDF) or others.

- The threshold $t$ seems to be a critical hyper-parameter. The default value 1e-7 seems to be tricky. The authors should provide more details on how t was chosen and whether its value was tuned experimentally or derived theoretically. Additionally, discussing the sensitivity of the model's performance to changes in t would be insightful.

- It would be helpful to visualize the words filtered, and change of data category proportion before and after WFPP.

- How about the potential of WFPP on (vision-text) large language models?

---

> ### Author Response · Authors · 2024-11-25
>
> Dear Reviewer,
>
> Thank you for your thorough review of our submission. Please also check our overall answer. Here are our answers to your specific questions:
>
> Question 1: The paper uses a straightforward approach to calculate word frequency, which is essential for the WFPP method. It would be beneficial if the authors could discuss the choice of this particular frequency calculation over other potential methods, such as term frequency-inverse document frequency (TF-IDF) or others.
>
> In the revised paper, we have added additional discussion of the choice. In a nutshell, we made this choice because term frequency is directly understandable and easy to calculate. We plan to explore other options, such as TF-IDF, in future work.
>
> Question 2: The threshold seems to be a critical hyper-parameter. The default value 1e-7 seems to be tricky. The authors should provide more details on how t was chosen and whether its value was tuned experimentally or derived theoretically. Additionally, discussing the sensitivity of the model's performance to changes in t would be insightful.
>
> In the revised paper, we added an explanation of how the threshold was set. There’s no trick. We simply chose a threshold in a practical manner: When we set $t = 1e-7$, all words that occur < 20 times, $P(w_i)$ will be set to 1, which we use in our paper.  We did not optimize $t$ since our approach is not particularly sensitive to the value of $t$. In the revised paper, we explicitly state how the threshold is set (Section A.2) and also include the results of new experiments that show that $t = 1e-6$ (words occurring < 206 times, $P(w_i)$ set to 1) and $t = 1e-8$ (words occurring < 5 times, $P(w_i)$ set to 1) perform comparably.  In the revised paper, we have added a new table (Table 16) that shows that the performance of different thresholds are comparable, it just needs to be set roughly so that it’s reasonable.
>
> Question 3: It would be helpful to visualize the words filtered, and change of data category proportion before and after WFPP.
>
>  We added more information to Figure 2 to visualize the words that were filtered and a new table (Table 19) to provide information on how the categories change. Initially, Figure 2 displayed the distribution of the top 300 words before and after applying WFPP, but this format was not user-friendly for check word types. Therefore, we have updated Figure 2 and calculated the proportions of data categories before and after applying WFPP. Compared to the random method, WFPP removes more high-frequency words. We also analyze the percentages and counts for both random and frequency sampling methods. In addition, we visualize the word distribution for the first and second part of the data after applying WFPP, as shown in Figure 5. The second part has many more high-frequency words than the first part and the high-frequency words percentage of the second part is substantially higher than 50\%. You can see the details in Figures 2, 4, 5, and  Table 19 in the new version of our paper.
>
> Question 4: How about the potential of WFPP on (vision-text) large language models?
>
> This is an interesting, but non-trivial question. Intuitively a different loss, such as a masking loss of an LLM, might benefit from retaining a different type of data. We believe that WFPP has an overall data balancing effect that would be beneficial to other models. However, intuitively, a different loss, such as a masking loss of an LLM, might benefit from retaining a different type of data. For this reason, we conclude that to answer this question requires a careful, comprehensive empirical verification. We have considered it out of scope for the current paper.

---

> > ### Comment · Reviewer_TRgD · 2024-12-01
> >
> > Thank you for the authors' response and the additional helpful experiments. However, I still have concerns regarding the proposed method, particularly the fact that the probability of discarding samples is dependent on the input text length. This can be challenging, especially for pretraining samples with varying lengths, which is particularly the case for pre-training samples of LLMs nowadays. This issue, however, was not addressed in the response. Furthermore, the overall approach does not seem technically new. Given the aforementioned reasons, I will maintain my score.

---

> ### Author Response · Authors · 2024-12-03
>
> Dear Reviewer,
>
> Thank you for your comment. We point out that in Section, 4.5 "Impact of Text Length and Text Length Normalization" of the original submission we empirically demonstrated that retaining only longer texts deteriorates performance. On this basis, we conclude that it is appropriate that text-image pair pruning is length dependent and that it normalizes long texts. The novelty of our work is quite simply the lack of light-weight pruning methods for CLIP.

---

### Author Response · Authors · 2024-11-25

Dear reviewers and ACs,

Thank you for your thorough review of our submission. We greatly appreciate your constructive feedback and would like to address the weaknesses you've identified.

Strengths:

 We are pleased that the reviewers appreciate our method's simplicity and practicality (**Reviewer** **TRgD,  VHZe, eLe2, zmLE**). The reviewers have also recognized that it consistently outperforms existing models and other pruning methods across various benchmarks through comprehensive experiments (**Reviewer** **TRgD,  VHZe, eLe2)**. Further, the reviewers comment that the paper is well-written, clearly structured, and includes helpful visual aids like figures and tables, making the content accessible and easy to follow (**Reviewer** **TRgD,  VHZe, zmLE**). We have made all code and data publicly available to ensure reproducibility.

Weaknesses:

The CLIP model that we used in our experiments was pre-trained on CC3M and CC12M datasets, and does not achieve the same performance of CLIP models trained on larger data sets, such as the original CLIP[8] and MetaCLIP [1] (**Reviewer** **TRgD, eLe2**). Due to limited computing resources, we cannot train CLIP on large-scale datasets like LAION-400M. Based on WFPP's successful application in CLIP with the CC3M and CC12M datasets, we believe that WFPP can also apply on large-scale datasets.

Additionally, the paper needs more comprehensive citations and discussion of related work in efficient CLIP training and data pruning (**Reviewer eLe2, zmLE**).  We've added your recommended related works to cover different efficient CLIP training methods and data pruning approaches in NLP and CV.

While our experimental comparisons focus primarily on the original CLIP and MetaCLIP models, our data pruning method is complementary to other efficient CLIP training methods, such as DeCLIP [2], Filip [3], Uniclip [4], TinyCLIP [5], MobileCLIP [6], and ALIP [7] (**Reviewer eLe2)**. WFPP can be applied to these approaches to further improve their training efficiency.  Given the time limit of rebutal, It’s impossible to evaluate our method on them one by one. Instead, we provide an analysis on the additional datasets, COCO and VG, used in other method, where we show that the word distribution after applying WFPP is similar to the word distribution CC12M, which prune the text that inculde high frequency words to create a balance subset. Experimental results on additional data sets are shown in Figure 4,5,6 and 7 in our revised paper.

Moreover, we made this additions to the revised paper that we have uploaded in response to the reviewer comments: We conducted additional experiments to demonstrate our method's effectiveness with synthetic captions (**Question from** **Reviewer VHZe**). We analyzed threshold selection (**Question from** **Reviewer TRgD**) and performed detailed word and token analysis (**Question from Reviewer VHZe**). We also carried out data analysis to show the potential for our method can be applied across different datasets and models (**Question from** **Reviewer TRgD**).

We trust this response addresses your concerns and clarifies our work. We appreciate your feedback and look forward to further improving our paper based on your suggestions.

Best regards,

Authors

[1]  Demystifying clip data. ICLR 2024

[2] Supervision exists everywhere: A data-efficient contrastive language-image pre-training paradigm. ICLR 2022

[3] Filip: fine-grained interactive language-image pre-training. ICLR 2022

[4] Uniclip: Unified framework for contrastive language-image pretraining. NeurIPS 2022

[5] Tinyclip: Clip distillation via affinity mimicking and weight inheritance. ICCV 2023

[6] Mobileclip: Fast image-text models through multi-modal reinforced training. CVPR 2024

[7] Alip: Adaptive language-image pre-training with synthetic caption. ICCV 2023

[8] Learning Transferable Visual Models From Natural Language Supervision. ICML 2021

---

### Meta-Review · Area_Chair_xmKs · 2024-12-25

**Metareview:**

This paper proposes WFPP, a word-frequency-based pruning method for improving CLIP pre-training by removing image-text pairs containing high-frequency words. While the paper demonstrates some empirical improvements on CC3M/CC12M datasets and offers a simple, practical approach, several critical weaknesses make it unsuitable for acceptance. The key limitations include: (1) All experiments are conducted only on small datasets (CC3M/CC12M) with performance far below state-of-the-art models trained on larger datasets like LAION-400M, raising serious questions about scalability. (2) The theoretical foundation is weak - there's no clear explanation for why pruning based on word frequency should help with image-text alignment. (3) The method has concerning technical issues, like discarding probability being dependent on text length in an unintuitive way. (4) The experimental evaluation is limited - comparisons focus mainly on basic CLIP rather than more recent efficient training methods. Though the authors added some new experiments during rebuttal (e.g., with DeCLIP and synthetic captions), fundamental questions about scalability to large datasets remain unaddressed. Thus I vote to reject the paper.

**Additional Comments On Reviewer Discussion:**

The reviewers raised several significant concerns that sparked substantial discussion. Reviewer TRgD highlighted issues with text length dependency and questioned the technical novelty - while the authors pointed to empirical results showing benefits of length normalization, TRgD remained unconvinced about the fundamental approach. Reviewer VHZe asked about compatibility with synthetic captions and alternative metrics - the authors added new experiments showing WFPP can complement synthetic captions. Reviewer eLe2 requested comparisons with efficient CLIP training methods - the authors added DeCLIP experiments showing benefits of WFPP, leading eLe2 to increase their score. Reviewer zmLE criticized limited scope and baselines - while the authors added related work discussion, zmLE felt core concerns remained unaddressed and lowered their score. The rebuttal revealed a key divide - while WFPP shows empirical benefits in limited settings and can complement other methods, questions about theoretical justification and large-scale applicability remain open

---

### Decision · Program_Chairs · 2025-01-22

Reject